# Neural Network Out-of-Distribution Detection for Regression Tasks

## Abstract

Neural network out-of-distribution (OOD) detection aims to identify when a model is unable to generalize to new inputs, either due to covariate shift or anomalous data. Most existing OOD methods only apply to classification tasks, as they assume a discrete set of possible predictions. In this paper, we propose a method for neural network OOD detection that can be applied to regression problems. We demonstrate that the hidden features for in-distribution data can be described by a highly concentrated, low dimensional distribution. Therefore, we can model these in-distribution features with an extremely simple generative model, such as a Gaussian mixture model (GMM) with 4 or fewer components. We demonstrate on several real-world benchmark data sets that GMM-based feature detection achieves state-of-the-art OOD detection results on several regression tasks. Moreover, this approach is simple to implement and computationally efficient.

## 1 Introduction

The success of deep neural networks in many domains (Krizhevsky et al., 2012; Lample et al., 2016; Mnih et al., 2016) is due to their ability to learn complex functions that generalize to new samples. However, this observed generalization only extends to data that are sufficiently similar to the training data. If the neural network encounters data that deviates from the distribution of training data, its predictions are likely to be erroneous or nonsensical (Guo et al., 2017; Jiang et al., 2012; Begoli et al., 2019). This may occur if the model is used in scenarios that experience covariate shift (Sugiyama et al., 2007) or if the model encounters previously-unseen categories of data (Yu et al., 2017; Hassen & Chan, 2018). Such scenarios are examples of *out-of-distribution* (OOD) inputs.

Ideally we would like for neural networks to adapt to such shifts in the data distribution (Amodei et al., 2016). In the absence of such adaptation, *out-of-distribution detection* should be used to identify when a model is unable to generalize to a previously-unseen input. While there are several proposed methods for neural network OOD detection (Hendrycks & Gimpel, 2017; Liang et al., 2018; Lee et al., 2018b), many of these methods rely on architectural components specific to classification neural networks. Consequentially, they cannot be applied to regression problems. Regression neural networks typically output only a point prediction rather than a predictive distribution, and thus the output does not indicate its uncertainty or reliability for a given input. This is illustrated in Figure 1, which displays predictions from a network trained to predict prices of middle-class houses in Kentucky. The network outputs a house price prediction for any possible input – even for out-of-distribution images like a California mansion or chair. These predictions fall within the normal range of possible prices, and therefore do not convey that the inputs are not valid for this model.

Because the predictions cannot identify OOD inputs, we must look for alternative signals. Previous approaches perform regression OOD detection through ensembles (Gal & Ghahramani, 2016; Lakshminarayanan et al., 2017) or through an additional uncertainty prediction layer (Kendall & Gal, 2017; Malinin et al., 2017). In this paper, we instead turn to *the space of hidden features*. During training a neural network learns to extract relevant features about the training data and discards irrelevant information. Whether or not a network generalizes to a given test sample depends on the extracted features from that sample. Networks do not generalize to out-of-distribution data because the distributional shift causes the network to extract the wrong information. For example, when the housing neural network is applied to the California mansion in Figure 1, the network's features do not extract the relevant information that would indicate the true price of the house. (e.g. presence of

Kentucky House
(In-dist. Input)

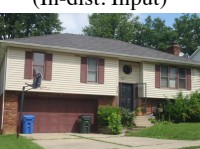

Actual: $130,000
Pred: **$132,000**
OOD Score: **0.05**

Kentucky House
(In-dist. input)

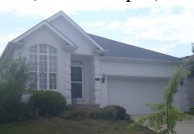

Actual: $125,000
Pred: **$156,000**
OOD Score: **0.11**

California Mansion
(Out-of-dist. input)

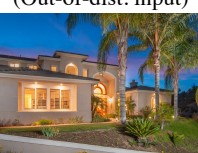

Actual: $1,430,000
Pred: **$270,000**
OOD Score: **0.86**

Chair
(Out-of-dist. input)

Actual: N/A
Pred: **$140,000**
OOD Score: **0.99**

Figure 1: A neural network trained to predict house prices in Kentucky from curb-side images. Although the algorithm generalizes well to within-distribution test images (left two houses), it has high error for (out-of-distribution) mansion from California and is forced to make a nonsensical prediction for the armchair. Both latter cases are detected as out-of-sample with high OOD scores.

palm trees). In other words, the extracted features of the California mansion will differ significantly from training data features and therefore the network does not generalize.

Based on this intuition, we investigate how to utilize the hidden features of regression neural networks for OOD detection. In particular, we make several contributions. First, we argue empirically and theoretically that we cannot identify OOD inputs simply using only the networks' predictions. However, we then demonstrate that the features of in-distribution inputs lie on an intrinsically low-dimensional portion of the feature space. It is unlikely that OOD inputs map to similar locations in feature space because of this low-dimensionality. We additionally show that, because of this low-dimensionality, it is possible to model the in-distribution features with simple generative models.

To evaluate our proposed approach, we develop new OOD detection tasks based on large-scale computer vision regression datasets. We evaluate the OOD detection performance of several generative models trained on in-distribution features. Surprisingly, we find that a simple mixture of Gaussians – often with no more than 2 components – is better at OOD detection than more complex models such as variational autoencoders. Finally, we demonstrate that GMM models of in-distribution features are able to outperform other regression OOD methods across several benchmarks.

## 2 RELATED WORK

Out-of-distribution detection and the related problems of outlier detection (Hodge & Austin, 2004; Chalapathy & Chawla, 2019) and novelty detection (Pimentel et al., 2014) are well-studied in statistics and machine learning. Arguably, the most straightforward method to these problems is generatively modeling the distribution of inputs $p(\mathbf{x})$ using a parametric distribution (Chow, 1970; Eskin, 2000) or a nonparametric density estimate (Kim & Scott, 2012). If $p(\mathbf{x})$ is small for a given input, then it is likely out-of-distribution or an anomaly. Several recent works (Choi et al., 2018; Nalisnick et al., 2019; Pidhorskyi et al., 2018) have suggested identifying OOD inputs with deep generative models (Goodfellow et al., 2014; Kingma & Welling, 2014; Rezende et al., 2014; Van Den Oord et al., 2016). One challenge with these methods is that deep generative models cannot necessarily model large-scale images or other complex distributions (Hendrycks et al., 2019) and may be overconfident when modeling input data (Nalisnick et al., 2019).

Rather than directly modeling the input distribution, the approach proposed in this paper operates on feature spaces using a mixture model to capture potentially anomalous inputs. This falls under the category of *model-dependent* OOD detection methods, which are ideal when the input distribution is too complex to model with generative methods (e.g. for high resolution images) (Hendrycks et al., 2019). Several model-dependent methods have been proposed for classification neural networks. The training procedure of a classification network can be modified to include uncertain samples (Lee et al., 2018a) or to discourage overconfidence through an alternative loss function (Alemi et al., 2018; Masana et al., 2018; Subramanya et al., 2017; Sensoy et al., 2018). Alternatively, OOD metrics can be constructed for classification based on the softmax probability output (Hendrycks et al., 2019; Liang et al., 2018) or related uncertainty scores (Schulam & Saria, 2019). In a similar vein to our proposed approach, (Lee et al., 2018b) use a class-conditional Mahalanobis distance

for classification OOD. All of these methods are designed specifically for multi-class classification problems – typically relying on the discrete output space. Consequentially, there are typically no straightforward extensions of these methods to regression problems.

There are fewer proposed methods for OOD detection on regression tasks. Most existing approaches rely on an estimate of the neural network's predictive uncertainty (Kuleshov et al., 2018). There are several proposed approaches to obtain uncertainty estimates from regression neural networks. One class of approaches is to add an additional output that predicts a confidence interval for the network's prediction (Lakshminarayanan et al., 2017; Kendall & Gal, 2017; Malinin et al., 2017). This layer can be trained in conjunction with the predictive output to minimize the negative log likelihood on the training set. Alternatively, Bayesian approaches to deep learning (Blundell et al., 2015; Gal & Ghahramani, 2016; Gal et al., 2017; Kingma et al., 2015) can be used to estimate predictive uncertainty on regression tasks. These approaches typically approximate the posterior distribution of neural networks parameters through an ensemble of models. Non-Bayesian ensembling approaches achieve can produce similar uncertainty estimates (Lakshminarayanan et al., 2017; Maddox et al., 2019). In this paper we propose an orthogonal approach to regression OOD detection – using the distribution of hidden features rather than the model's predictive uncertainty.

## 3    CHARACTERIZING OOD PREDICTIONS ON REGRESSION TASKS

In the classification setting, the output of a neural network is a softmax score for each class. Because this output can be interpreted as a probability distribution over classes, it represents some notion of the neural network's uncertainty and therefore can be used to identify inputs that are likely to be OOD (Hendrycks & Gimpel, 2017; Liang et al., 2018). For most regression architectures however, the output is typically a point prediction $\hat{y}$ which does not convey any notion of predictive uncertainty. Moreover, in this section we show that these point predictions $\hat{y}$ are insufficient for differentiating between in-distribution and out-of-distribution inputs.

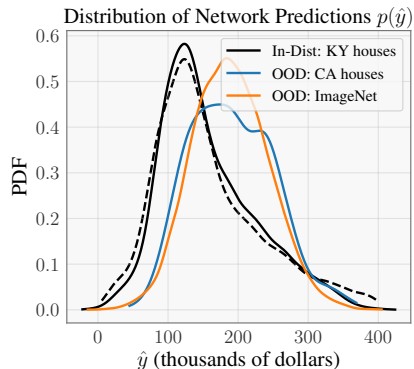

To illustrate this, we train a neural network to predict the price of homes in Kentucky based on a front facing picture. Figure 2 (left) displays a histogram of the network's predictions $\hat{y}$ on a withheld validation set (black line). The distribution of these predictions corresponds to the distribution of actual house prices (dotted line). However, if the same network receives an OOD input, it predicts price values that lie in the same range as the in-distribution predictions. For example, if houses from California are input into the neural network, the predicted housing prices (blue line) are roughly the same as that of Kentucky houses despite these houses costing up to 10 times as much. The network predicts similar values for Imagenet images (orange line) even though these images do not contain any houses. Based on the $\hat{y}$ alone, it is not possible to determine that these inputs are not valid.

Figure 2:  A neural network is trained to predict Kentucky housing prices. The distribution of predictions on the validation data (black line) and the actual distribution of housing prices (dotted line). Out-of-distribution data (blue and green lines) obtain similar predictions.

The fact that the OOD predictions are similar to in-distribution predictions can be explained by a simple curse-of-dimensionality argument. Let $\phi$ be a regression neural network that maps an input $\mathbf{x} \in \mathcal{X}$ to an output $\hat{y} \in \mathbb{R}$. The last layer of $\phi$ produces a set of activations $\mathbf{h}(\mathbf{x})$ that are linearly mapped to the one-dimensional prediction: $\hat{y} = \mathbf{w}^\top \mathbf{h}(\mathbf{x})$. Let's assume that the features $\mathbf{h}(\mathbf{x})$ for some OOD inputs can be bounded within some Euclidean ball of radius $r$. If the dimensionality of $\mathbf{h}(\mathbf{x})$ is large, then the probability of an extreme prediction becomes exponentially unlikely.

**Remark 1.** *Let $\mathbf{h}(\mathbf{x})$ be features drawn from a uniform measure on a Euclidean ball of radius $r$. Let $d$ be the dimensionality of $\mathbf{h}(\mathbf{x})$. The probability of a large prediction $|\hat{y}| > \epsilon$ is bounded by*

$$p(|\hat{y}| > \epsilon) \leq \frac{2\Gamma(d/2 + 1)}{\pi^{d/2} r^d} \exp\left(\frac{-d\epsilon^2}{2r^2 \|\mathbf{w}\|_2^2}\right).$$

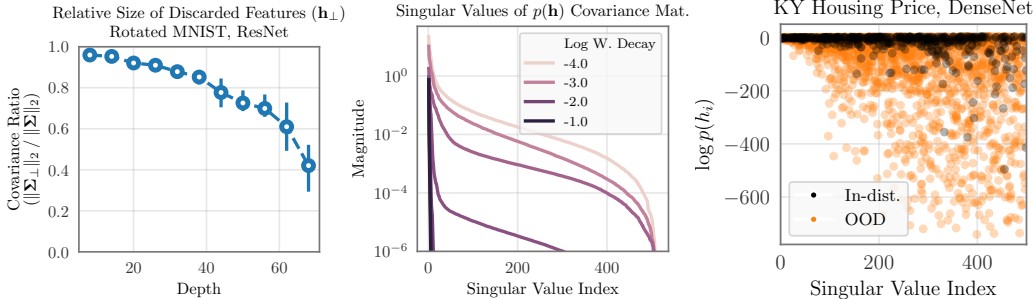

Figure 3: **Left:** The relative size of "discarded" feature information $\mathbf{h}_\perp(\mathbf{x})$ decreases as the depth of neural networks increases. **Center:** The intrinsic dimensionality of the feature vectors $\mathbf{h}(\mathbf{x})$, as measured by spectrum of its principal components, decreases with weight decay. **Right:** Likelihood of in-distribution and OOD features as measured by a Gaussian decomposed into the likelihood of each principal component. The Gaussian was fit on (in-distribution) training set features. OOD features have low likelihood on the small principal components, suggesting that they do not lie in the same low-dimensional distribution.

(See Appendix B for proof). In practice it is not necessarily the case that OOD features will be uniformly distributed. Nevertheless, this remark illustrates that predictive outputs discard information that would indicate whether a sample is OOD or not.

## 4 CHARACTERIZING THE DISTRIBUTION OF HIDDEN FEATURES

Although the network's predictions look very similar for in-distribution and OOD inputs, we expect that their hidden features $\mathbf{h}(\mathbf{x})$ are very different. Since ImageNet pictures do not contain front-facing houses, the network will extract very different types of features. We similarly expect the activations of California houses to be different since the network is unable to generalize these inputs. To that end, we hypothesize that a generative model of in-distribution features $p(\mathbf{h}(\mathbf{x}))$ will distinguish between in- and out-of-distribution inputs. In this section we demonstrate that – despite the high dimensionality of $\mathbf{h}(\mathbf{x})$ – the in-distribution features are very tightly concentrated. As a result, it is possible to model in-distribution features with a simple generative model.

**In-distribution features are intrinsically low dimensional.** In the regression setting, the network's prediction is the inner product between the features and a weight vector: $\hat{y} = \mathbf{w}^\top \mathbf{h}(\mathbf{x})$. Any feature information that is orthogonal to the weight $\mathbf{w}$ will not contribute to the prediction and therefore is "discarded" information. We can describe a feature vector $\mathbf{h}(\mathbf{x})$ in terms of the prediction $\hat{y}$ and the discarded information $\mathbf{h}_\perp(\mathbf{x})$:

$$\mathbf{h}(\mathbf{x}) = \hat{y}\mathbf{w} + \mathbf{h}_\perp(\mathbf{x}), \qquad \mathbf{h}_\perp(\mathbf{x}) = \left(\mathbf{I} - \mathbf{w}\mathbf{w}^\top\right)\mathbf{h}(\mathbf{x}),$$

where $\mathbf{h}_\perp(\mathbf{x})$ is the discarded information, computed using the orthogonal projection of $\mathbf{w}$. Also, we assume, without loss of generality, that $\|\mathbf{w}\|_2 = 1$. It is worth emphasizing that "useful" predictive information $\hat{y}\mathbf{w}$ lies within a single dimension of the $d$-dimensional $\mathbf{h}(\mathbf{x})$ vector (where $d$ is often greater than 500). Though the "discarded" information $\mathbf{h}_\perp(\mathbf{x})$ occupies $d-1$ dimensions, it typically has low intrinsic dimensionality for in-distribution data as we demonstrate below. Consequentially, the in-distribution features $\mathbf{h}(\mathbf{x}) = \mathbf{h}_\perp(\mathbf{x}) + \hat{y}\mathbf{w}$ can be described by a low-dimensional distribution. We identify two factors that are correlated with this low dimensionality: network capacity and regularization.

**Network capacity reduces the relative size of $\mathbf{h}_\perp(\mathbf{x})$.** As network capacity increases, we expect that there will be less "discarded" information $\mathbf{h}_\perp(\mathbf{x})$. To understand why this is the case, assume there is a $l$-layer network in which there is some correlation between the discarded features and the true labels. Now assume we add an additional layer before the linear layer which computes a new set of hidden features $\mathbf{h}'(\mathbf{x})$. This new layer can improve the network's predictions if it uses information in $\mathbf{h}_\perp(\mathbf{x})$ to augment the original prediction $\hat{y}$. Consequentially, this new network will likely have less discarded information than the shallower network.

While this is just an illustrative example, we can empirically demonstrate that network capacity does in fact decrease the amount of discarded information. First, we introduce the following metric of discarded information: let $\Sigma$ be the empirical covariance matrix of $\mathbf{h}(\mathbf{x})$ for in-distribution data, and let $\Sigma_\perp$ be the covariance of $\mathbf{h}_\perp(\mathbf{x})$. We measure the amount of discarded information by comparing the maximum singular values of $\Sigma$ and $\Sigma_\perp$. If $\|\Sigma_\perp\|_2/\|\Sigma\|_2$ is close to 1, then the feature vectors are comprised mostly of discarded information. Conversely, $\mathbf{h}(\mathbf{x})$ will consist primarily of predictive information if $\|\Sigma_\perp\|_2/\|\Sigma\|_2$ is close to zero.

We apply this metric to a simple regression task using the MNIST dataset. Each input image is rotated by an amount between $-90$ and $90$ degrees. We train residual networks with increasing depth to predict the amount of rotation in each image (details in Appendix C). Figure 3 (left) plots the singular value ratio $\|\Sigma_\perp\|_2/\|\Sigma\|_2$ as a function of network depth. The ratio is quite large for shallow networks, but decreases with depth. This suggests that, for deep networks, most of $\mathbf{h}(\mathbf{x})$ lies on the one-dimensional subspace defined by the weight vector $\mathbf{w}$.

**Regularization reduces the dimensionality of $\mathbf{h}(\mathbf{x})$.** Applying weight decay to a neural network reduces the magnitude of the weights, which in turn reduces the magnitude of feature vectors. Reducing the magnitude of $\mathbf{h}(\mathbf{x})$ while maintaining predictive power will result in feature vectors with little discarded information $\mathbf{h}(\mathbf{x})$. Since all but one dimensions of the feature space are discarded, we expect that the intrinsic dimensionality of features will decrease with increasing regularization. To support this intuition we train the LeNet model on rotated MNIST with various amounts of weight decay. In Figure 3 (center) we plot the spectrum of the in-distribution feature covariance matrix $\Sigma$ for these different models. We see that all covariance matrices are dominated by a few large singular values. However, the spectra of the high regularization models decay much more rapidly, suggesting that regularization simplifies the distribution of hidden features.

The fact that in-distribution inputs occupy a low-dimensional portion of high-dimensional feature space is extremely advantageous. If we draw a random set of random features $\mathbf{h}(\mathbf{x})$ (e.g. from a uniform measure on a Euclidean ball), it will be highly unlikely that these features occupy the same low-dimensional space. Though OOD features are not truly random vectors in practice, we find empirically that they do not occupy the same low dimensional subspace. To demonstrate this, we fit a multivariate Gaussian to the training set features from the housing price network. We can decompose the fit of this Gaussian as a product of its principle components and measure each component's likelihood for in- and out-of-distribution data. Figure 3 (right) displays the fits for the housing price validation data (in-distribution) and ImageNet samples (OOD). We find that in-distribution and OOD data are equally likely for the main principal components. However, OOD features do not fit the smaller principal components well, and thus these small principal components are able to distinguish in- and out-of-distribution data. In this case the curse-of-dimensionality makes it possible to distinguish OOD and in-distribution features.

## 5   OOD DETECTION WITH GENERATIVE MODELS OF HIDDEN FEATURES

As we demonstrate in the previous section, the distribution of in-distribution hidden features $\mathbf{h}(\mathbf{x})$ is intrinsically low dimensional. Since OOD data are unlikely to share the same low-dimensional distribution, we propose using generative models of in-distribution $\mathbf{h}(\mathbf{x})$ to identify OOD inputs.

**Possible generative model.** While a simple Gaussian can model the distribution $\mathbf{h}(\mathbf{x})$, it is possible that a more complex generative model will fit better. There are several choices of generative models, arguably the simplest choice are parametric models such as Gaussian mixture models (**GMMs**), which can approximate any distribution with enough mixture components Bishop (2006). Nonparameteric models, such as kernel density estimation (**KDE**), offer complexity that scales with the number of observations. Deep generative models such as variational autoencoders (**VAE**) (Kingma & Welling, 2014; Rezende et al., 2014) are arguably the most powerful class of generative models.

**Model fitting.** After training the predictive neural network, a generative model is fit to in-distribution features. In all experiments, we use features extracted from the neural network's training data. However, we note that features from any (unlabeled) in-distribution data could be used instead.

**Model selection and hyperparameter tuning.** Tuning hyperparameters can be challenging because one typically does not have access to OOD data. For our method we must determine 1) which generative model to use and 2) the hyperparameters of the generative model. We select these hyperparameters using the generative model's log-likelihood on the validation set: $\log p(\mathcal{D}_{\text{valid}}) = \sum_{\mathbf{x}} \log p(\mathbf{h}(\mathbf{x}))$. From a decision theoretic standpoint, the model with the largest likelihood fits the validation data with the smallest distribution (Bishop, 2006). If the distribution is strongly peaked then it has less support for samples that are not in-distribution, and therefore will be good for OOD detection. Importantly, this criterion only requires access to in-distribution data.

**Comparing the Performance of Generative Models.** In Section A.1 we examine how the choice of generative model affects OOD detection performance. We compare KDE models, VAEs, and 2-component GMMs on several benchmark OOD tasks (which are described in detail in the next section). Surprisingly, we find that GMMs detects OOD inputs as well – if not better – than the other models. This is especially surprising because GMMs tend not to perform well in high dimensional spaces (Bishop, 2006), which further suggests that in-distribution features are highly concentrated.

## 6 Experiments

In this section, we demonstrate the performance of our OOD detection method on several regression tasks. We first illustrate the method on a 2D toy example, and then we evaluate the method on two large-scale computer vision regression tasks. For experimental details, we refer the reader to Appendix C. Based on the ablation study in Section A.1, we model the in-distribution features with a GMM. The number of components in the GMM is chosen through model selection with the validation dataset. (The effect of the GMM size is explored in Section A.2.)

**Smoothed XOR.** To illustrate how feature spaces can be used for detecting OOD samples, we consider a simulated regression task. The goal is to approximate the smoothed XOR function from noisy labeled data. In-distribution inputs consist of Uniform samples in $[0,1]^2$, while OOD inputs come from an isotropic Gaussian distribution centered at $[1.5, 1.5]$ with variance $\sigma^2 = 0.1$. Figure 4 (left) depicts the proposed setting, i.e. the target XOR function and in-dist./OOD samples. Here, we use multilayer perceptrons (MLP) as the predictive model. Figure 4 (middle) plots predictions for both OOD and in-distribution (validation) samples. Note that this corresponds to the desired scenario in which one cannot identify OOD inputs based on the predictions alone.

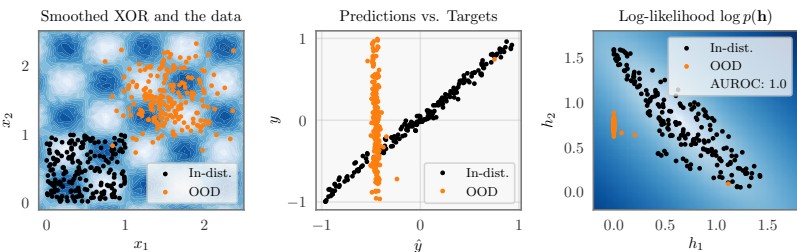

Figure 4: Smoothed XOR regression task. **Left**: Target XOR function over in-dist. and OOD data. **Middle**: Predictions for in-dist. and OOD inputs. **Right**: Gaussian fit on the in- and OOD features.

The proposed approach relies on the assumption that hidden features are informative, and their distribution can be approximated using simple generative models. In this regard, Figure 4 (right) shows the fit from a Gaussian on the activations at the last nonlinear layer of the prediction network. There is virtually no overlap between in-distribution and OOD features. As a consequence, we achieve perfect detection, as represented by an *area under the ROC curve* (AUROC) equals to 1.

**Housing price prediction.** We demonstrate our proposed method on a large-scale regression task: predicting home prices from front-facing house pictures. We train a 121-layer DenseNet (Huang et al., 2017) and a 50-layer ResNet (He et al., 2016) on a dataset of nearly 50,000 middle-income

Table 1: OOD detection results for housing price prediction (top) and age estimation (bottom).

| Network | OOD Dataset | TNR at TPR 95% | | | | | AUROC | | | | |
|---|---|---|---|---|---|---|---|---|---|---|---|
| | | MCD | Ens. | Var. | Var. + MCD | GMM | MCD | Ens. | Var. | Var. + MCD | GMM |
| KY Housing Price (DenseNet-121) | ImageNet | 0.223 | 0.537 | 0.757 | 0.200 | **0.999** | 0.825 | 0.909 | 0.940 | 0.764 | **0.999** |
| | KITTI | 0.475 | 0.517 | 0.443 | 0.318 | **0.994** | 0.910 | 0.917 | 0.910 | 0.830 | **0.995** |
| | StreetView | 0.280 | 0.395 | 0.348 | 0.238 | **0.936** | 0.847 | 0.877 | 0.849 | 0.779 | **0.982** |
| | CA Houses | 0.282 | 0.304 | 0.454 | 0.207 | **0.568** | 0.832 | 0.832 | 0.874 | 0.757 | **0.928** |
| KY Housing Price (ResNet-50) | ImageNet | 0.199 | 0.711 | 0.629 | 0.611 | **0.998** | 0.799 | 0.953 | 0.931 | 0.928 | **0.999** |
| | KITTI | 0.518 | 0.731 | 0.691 | 0.687 | **0.932** | 0.922 | 0.961 | 0.954 | 0.955 | **0.983** |
| | StreetView | 0.317 | 0.680 | 0.609 | 0.603 | **0.868** | 0.833 | 0.951 | 0.933 | 0.933 | **0.975** |
| | CA Houses | 0.224 | **0.598** | 0.261 | 0.263 | 0.574 | 0.803 | **0.930** | 0.839 | 0.842 | 0.924 |
| Adult Age Estimation (DenseNet-121) | ImageNet | 0.256 | 0.601 | 0.908 | 0.241 | **0.983** | 0.799 | 0.903 | 0.981 | 0.760 | **0.994** |
| | Streetview | 0.420 | 0.625 | 0.963 | 0.361 | **0.984** | 0.873 | 0.916 | 0.981 | 0.809 | **0.995** |
| | Pedestrian | 0.590 | 0.794 | 0.998 | 0.313 | **1.000** | 0.925 | 0.957 | **0.999** | 0.804 | **0.999** |
| | Child Age | 0.189 | 0.201 | 0.338 | 0.125 | **0.477** | 0.670 | 0.689 | 0.844 | 0.610 | **0.866** |
| Adult Age Estimation (ResNet-50) | ImageNet | 0.310 | 0.915 | 0.898 | 0.876 | **0.937** | 0.823 | 0.978 | 0.977 | 0.970 | **0.987** |
| | Streetview | 0.240 | 0.987 | **0.998** | 0.990 | 0.908 | 0.826 | 0.987 | **0.996** | 0.993 | 0.979 |
| | Pedestrian | 0.277 | 0.982 | **0.998** | 0.998 | 0.986 | 0.851 | 0.985 | **0.999** | 0.999 | 0.993 |
| | Child Age | 0.150 | 0.234 | 0.183 | 0.148 | **0.434** | 0.656 | 0.627 | 0.540 | 0.553 | **0.867** |

Figure 5: Housing price predictions – (**Left**) Prediction vs. actual price for in-distribution and OOD houses. (**Middle-left**) The OOD score $p(\mathbf{h}(\mathbf{x}))$ from the GMM correlates with the predictive error and can identify the most erroneous housing predictions. Age estimations – (**Middle-right**) Prediction vs. actual age for adults (in-distribution) and children (OOD). (**Right**) Similarly to housing, the GMM's OOD score identifies high-error predictions.

house images from Fayette County in Kentucky. The CNNs are pre-trained on ImageNet (Deng et al., 2009) and the last layer is replaced with a linear layer that produces a single output. We compare the GMM-based detector against methods that use ensemble or Bayesian model averages to estimate predictive uncertainty. **Monte-Carlo Dropout** (MCD) (Gal & Ghahramani, 2016; Kendall & Gal, 2017) creates a Bayesian ensemble of 10 neural networks, where each network is sampled from a base network with dropout ($p = 0.2$). **Deep Ensembles** (Ens.) (Lakshminarayanan et al., 2017) is an ensemble of 4 standard neural networks. For these two baselines, the variance of the forward passes is used as a metric for detecting OOD inputs. **Var.** (Malinin et al., 2017) adds a variance output to the neural network to predict uncertainty estimates. **Var. + MCD** (Kendall & Gal, 2017) combines this variance estimate with MC-Dropout.

Following Hendrycks & Gimpel (2017), we use several external datasets as sources for out-of-distribution inputs. Firstly, **ImageNet** (Deng et al., 2009) is a dataset of photos from 1,000 classes. This dataset contains easily detectable out-of-distribution inputs, since most of the pictures do not contain houses. **KITTI** (Geiger et al., 2013) is a dataset of self-driving car images. These images may contain some houses, but any houses will be in the periphery of the image rather than being centered. **StreetView** (Zamir & Shah, 2014) contains images from Google street view, which may also contain some frontal house views. The most challenging dataset is **CA Houses** (Ahmed & Moustafa, 2016), which consists of house images from California.

METRICS. We apply each detection method to the OOD datasets as well as a withheld test set of in-distribution images. The methods assign an OOD score to each input. Similar to Hendrycks & Gimpel (2017), we evaluate these scores based on two objectives: the true-negative detection rate at a 95% true-positive rate (TNR) and the area under the ROC curve (AUROC).

RESULTS. The results for housing price OOD detection are in Table 1 (first two sets of rows). We observe several trends. Firstly, the GMM detector substantially outperforms Monte-Carlo dropout and ensembles on the ImageNet, KITTI, and StreetView datasets. On ImageNet, the GMM detector achieves nearly perfect OOD detection rates. These datasets contain invalid inputs (i.e. inputs that do not contain houses), and therefore the out-of-distribution and in-distribution activations will be extremely different. The CA Houses dataset is the most challenging dataset for all of the detection methods. This is because this dataset represents covariate shifted data (i.e. inputs that contain different houses), and therefore, the OOD activations will be similar to in-distribution activations. Nevertheless, the GMM detector achieves a high TNR and AUROC that are comparable with the performance of an ensemble. In Figure 5 (far left) we plot the network's prediction and the actual housing price for in-distribution houses and CA houses (log scale). Because the CA houses are OOD, the predictions are over an order of magnitude different than the actual housing price. Figure 5 (middle-left) compares the error and the OOD score assigned by the GMM predictor. We see that there is a correlation between the predictive error and the OOD score. If we threshold the OOD score to obtain $95\%$ true-positive-rate, then the most erroneous samples will be labeled out-of-distribution.

**Age estimation.** We evaluate GMM-based OOD detection on another large-scale regression task: predicting a person's age from a portrait image. We take 20,000 images of people ages 20 and older from the UTKFace dataset (Zhang et al., 2017). 5,000 images are withheld for validation and testing. Similar to the previous setup, the CNNs are pre-trained on ImageNet and the last layer is replaced with a linear layer. We compare against the same baselines using the same metrics.

For out-of-distribution inputs, we use images from the **ImageNet** and **StreetView** datasets. While both datasets do contain some images of people, most of the images are not portraits and therefore are OOD. Additionally, we use 10,000 frames from the **Caltech Pedestrian** dataset (Dollár et al., 2012). These images, which were taken from vehicle driving through an urban environment, contain candid images of pedestrians. Finally, we generate a challenging OOD dataset using images of children 10 and younger from the UTKFace dataset (**Child**).

RESULTS. The results for age estimation are in Table 1 (last two sets of rows). Similar to our previous task, the GMMs achieve very high OOD detection performance on all datasets. The GMM detector for DenseNet-121 outperforms Monte-Carlo dropout and ensembles on all benchmarks by a significant margin. For ResNet-50, the GMM outperforms the other methods on all but one dataset with respect to both the TNR and AUROC metrics. The child age dataset is the most challenging OOD detection dataset for all methods. This is because the images are very similar to the training data. In Figure 5 (middle-right) we plot the network's prediction and the actual ages for adults (in-distribution) and children (OOD). The network is unable to correctly predict the ages of children. Figure 5 (right) shows that the GMM's score can be used to identify the most potentially erroneous predictions on this dataset.

**Other considerations.** Feature-based OOD detection is a computationally advantageous method. Using a GMM for OOD detection requires a constant number of matrix-vector multiplications for each input. This is a fraction of the cost of the neural network's prediction. It is worth noting that ensemble-based methods have benefits besides OOD detection, such as improved predictive performance. However, ensembles require multiple forward passes, which can be quite expensive.

## 7 CONCLUSION

Regression neural networks, unlike classification networks, output a point prediction rather than a distribution over possible predictions. From this prediction alone, we demonstrate that is nearly impossible to determine whether the network's input is anomalous or out-of-distribution. In this paper, we argue that the network's hidden features indicate whether samples are in-distribution or not. We demonstrate that the distribution of neural network features $p(\mathbf{h}(\mathbf{x}))$ are intrinsically low dimensional and can be well-approximated by a simple generative model, such as mixture of Gaussians. This is an simple approximation, especially considering that the feature space has more than $1,000$ dimensions. Nevertheless, we find that this approach is incredibly accurate at identifying OOD inputs, whether the inputs come from a nonsensical distribution or a slightly-shifted distribution.

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

# A  ABLATION STUDIES

## A.1  GENERATIVE MODELS

Here, we consider three methods for modeling hidden features: kernel density estimation (KDE), Gaussian mixture models (GMM), and variational autoencoders (VAE). The GMM consists of a 2-component mixture (2-GMM) with full covariance matrices. The hyper-parameters for KDE and VAE are selected using a withheld in-distribution validation set. The model that produces the highest average log-likelihood is selected.

For KDE, we consider gaussian kernels with widthbands in $\{10^{-5}, 10^{-4}, \dots, 10^{-1}\}$. Regarding VAEs, we train encoder and decoder networks with 3 and 4 hidden layers. The largest architecture consists of $(800, 512, 256, 128)$ hidden units. We have also considered simpler architectures with $(64, 32, 16)$. The networks are trained over 800 epochs using Adam with learning rate of $10^{-3}$. We also apply weight decay of $10^{-4}$ and batch normalization (batch size of 128).

Table 2 shows the performance of the generative models on housing price data. Overall, 2-GMM is the best performing model. For the easiest task (ImageNet), VAE and 2-GMM achieve very similar performances in terms of both TNR at TPR 95% and AUROC. Also, for all datasets the difference between VAE and 2-GMM is not higher than 2% AUROC.

On the other hand, KDE performs poorly compared to VAE and 2-GMM. A possible explanation for this erratic behavior lies in the fact that KDE is very sensitive to the choice of kernel bandwidth. Since it uses an isotropic kernel, a large bandwidth may lead to underfitting. It is worth emphasizing that by limiting the bandwidth to small values, KDE performance significantly increases, approximating those from VAE and GMM.

Table 2 (last two sets of rows) shows results on age estimation. Similar to the results on housing prices, GMM outperforms both KDE and VAE. The accuracy gap is notably higher for the RestNet network.

Table 2: Performance of generative models for OOD detection.

| Network | OOD Dataset | TNR at TPR 95% | | | AUROC | | |
|---|---|---|---|---|---|---|---|
| | | KDE | 2-GMM | VAE | KDE | 2-GMM | VAE |
| KY Housing Price (DenseNet-121) | ImageNet | 0.645 | **0.999** | 0.997 | 0.781 | **0.999** | **0.999** |
| | KITTI | 0.566 | **0.994** | 0.990 | 0.701 | **0.995** | 0.994 |
| | StreetView | 0.529 | **0.931** | 0.846 | 0.678 | **0.982** | 0.972 |
| | CA Houses | 0.387 | **0.553** | 0.550 | 0.640 | **0.923** | 0.915 |
| KY Housing Price (ResNet-50) | ImageNet | 0.705 | **0.996** | 0.994 | 0.833 | **0.999** | 0.998 |
| | KITTI | 0.530 | **0.908** | 0.878 | 0.688 | **0.981** | 0.977 |
| | StreetView | 0.485 | **0.820** | 0.722 | 0.653 | **0.968** | 0.957 |
| | CA Houses | 0.413 | **0.538** | 0.495 | 0.677 | **0.909** | 0.891 |
| Adult Age Estimation (DenseNet-121) | ImageNet | 0.723 | **0.983** | 0.955 | 0.840 | **0.994** | 0.990 |
| | Streetview | 0.734 | **0.984** | 0.957 | 0.839 | **0.995** | 0.987 |
| | Pedestrian | 0.854 | **1.000** | **1.000** | 0.911 | **0.999** | **0.999** |
| | Child Age | 0.297 | **0.477** | 0.456 | 0.642 | **0.866** | 0.854 |
| Adult Age Estimation (ResNet-50) | ImageNet | 0.393 | **0.845** | 0.664 | 0.725 | **0.974** | 0.927 |
| | Streetview | 0.297 | **0.758** | 0.513 | 0.721 | **0.958** | 0.899 |
| | Pedestrian | 0.365 | **0.925** | 0.753 | 0.744 | **0.982** | 0.959 |
| | Child Age | 0.336 | **0.475** | 0.471 | 0.662 | **0.878** | 0.873 |

## A.2  NUMBER OF GMM MIXTURE COMPONENTS.

If we fit the distribution of training set features with a GMM, then the only hyperparameter is the number of mixture components. We find that OOD detection performance is actually quite robust to this hyperparameter.

Table 3: OOD detection performance as a function of the GMM mixture size (1, 2, or 4). We display the TNR and AUROC for each GMM, as well as the GMM's log likelihood on in-distribution features. Blue columns correspond to the model with the highest log likelihood (the model selection criterion). Best results are in bold.

| Network | OOD Dataset | TNR at TPR 95% | | | AUROC | | |
|---|---|---|---|---|---|---|---|
| | | 1-GMM | 2-GMM | 4-GMM | 1-GMM | 2-GMM | 4-GMM |
| | *Log Lik. (In-dist.)* | 2924 | 2970 | **2991** | 2924 | 2970 | **2991** |
| KY Housing Price | ImageNet | 0.998 | **0.999** | **0.999** | **0.999** | **0.999** | **0.999** |
| (DenseNet-121) | KITTI | 0.992 | **0.994** | **0.994** | 0.994 | **0.995** | **0.995** |
| | StreetView | 0.914 | 0.931 | **0.936** | 0.980 | **0.982** | **0.982** |
| | CA Houses | 0.510 | 0.553 | **0.568** | 0.918 | 0.923 | **0.928** |
| | *Log. Lik. (In-dist.)* | 3864 | 3838 | **3903** | 3864 | 3838 | **3903** |
| KY Housing Price | ImageNet | 0.997 | 0.996 | **0.998** | **0.999** | **0.999** | **0.999** |
| (ResNet-50) | KITTI | **0.939** | 0.908 | 0.932 | **0.984** | 0.981 | 0.983 |
| | StreetView | 0.844 | 0.801 | **0.868** | 0.973 | 0.967 | **0.975** |
| | CA Houses | 0.482 | 0.538 | **0.574** | 0.904 | 0.908 | **0.924** |
| | *Log Lik. (In-dist.)* | 2563 | **2589** | 2588 | 2563 | **2589** | 2588 |
| Adult Age Estimation | ImageNet | **0.986** | 0.983 | 0.964 | **0.996** | 0.994 | 0.993 |
| (DenseNet-121) | Streetview | **0.989** | 0.984 | 0.982 | **0.997** | 0.995 | 0.994 |
| | Pedestrian | **1.000** | **1.000** | **1.000** | **1.000** | 0.999 | 0.999 |
| | Child Age | 0.416 | **0.477** | 0.427 | 0.852 | **0.866** | 0.860 |
| | *Log Lik. (In-dist.)* | **2630** | 2552 | 1876 | **2630** | 2552 | 1876 |
| Adult Age Estimation | ImageNet | **0.937** | 0.845 | 0.830 | **0.987** | 0.974 | 0.971 |
| (ResNet-50) | Streetview | **0.908** | 0.758 | 0.766 | **0.979** | 0.959 | 0.962 |
| | Pedestrian | **0.986** | 0.924 | 0.906 | **0.993** | 0.982 | 0.981 |
| | Child Age | 0.434 | **0.475** | 0.468 | 0.867 | 0.878 | **0.880** |

In Table 3 we display the OOD detection results for GMMs with 1, 2, and 4 mixture components. In addition, we report the log likelihood of the GMM on the in-distribution validation data, which we use as the selection criterion. The GMMs with the highest log likelihood for a given model are displayed in blue.

From this figure, we can observe a number of trends. Firstly, we notice that more mixture components does not always correspond to a better model. 4 mixture components achieves the highest log likelihood on the housing price networks; however, fewer components is better for age estimation. Nevertheless, the log likelihood functions as a good model selection criterion. The GMMs with the highest log likelihood tend to achieve the best OOD detection performance (or comparable to the best performance) both in terms of TNR and AUROC.

Surprisingly, we find the log likelihood is relatively robust to the number of mixture components. For example, the housing price ResNet's log likelihood values vary between 3838 and 3093. Similarly, the age estimation DenseNet's log likelihood values vary between 2563 and 2589. These similar likelihoods correspond to similar OOD detection performance. On all OOD detection benchmarks, the models achieve AUROCs that are within one percentage point of each other. The only experiment with highly varying log likelihood is the age estimation ResNet. The 4-component GMM obtains a significantly lower log likelihood than the 2- or 1-component mixtures. This low log likelihood corresponds to worse OOD detection performance. The 2- and 1-component mixtures perform significantly better on most OOD tasks than the 4-component mixture.

# B  PROOF OF REMARK 1

*Proof.* We assume that the features $\mathbf{h}(\mathbf{x})$ are drawn from a uniform measure on a Euclidean ball of radius $r$. The prediction corresponding to these features is given by $\hat{y} = \mathbf{w}^T \mathbf{h}(\mathbf{x})$, where $\mathbf{w}$ is the weight of the network's linear layer. In order for $\hat{y} > \epsilon$ for some $\epsilon$ it must be the case that $\mathbf{h}(\mathbf{x})$ lies

in the hyperspherical cap defined by the set of points $\mathbf{w}^T\mathbf{h}(\mathbf{x}) > \epsilon$, $\|\mathbf{h}(\mathbf{x})\|_2 \leq r$. I.e.

$$p(\hat{y} > \epsilon) = \frac{\text{Volume of cap}}{\text{Volume of ball}}.$$

The hypervolume of the Euclidean ball in a $d$ dimensional space is given by $\frac{\pi^{d/2}r^d}{\Gamma(d/2+1)}$, where $\Gamma$ represents the gamma function. While there is an exact formula for the for the hypervolume of a cap, we instead choose to use the more interpetable bound of Tkocz (2012):

$$\text{Volume of cap} < \exp\left(\frac{-d\epsilon^2}{2r^2\|\mathbf{w}\|_2^2}\right).$$

Finally, by symmetry we have that $p(\hat{y} > \epsilon) = p(\hat{y} < \epsilon)$. Putting this all together, we have that

$$p(|\hat{y}| > \epsilon) \leq \frac{2\Gamma(d/2+1)}{\pi^{d/2}r^d}\exp\left(\frac{-d\epsilon^2}{2r^2\|\mathbf{w}\|_2^2}\right).$$

$\square$

## C  EXPERIMENTAL DETAILS

### C.1  ROTATED MNIST EXPERIMENTS

The amount of rotation applied to each input is drawn from the distribution $\mathcal{N}(0, \pi/4)$. As a result, roughly 95% of the samples are rotated between $-\pi/2$ and $\pi/2$ degrees.

The ResNet architecture in the depth experiments is based on the architecture of He et al. (2016). The first convolutional layer is reduced to accept single-channel inputs, and the ReLU activations are replaced with `tanh` activations. The LeNet architecture follows that of LeCun et al. (1989). All networks are trained with SGD for 50 epochs, using an initial learning rate of 0.1 that is dropped by 10 after 25 and 37. We compute the covariances matrices $\mathbf{\Sigma}$ and $\mathbf{\Sigma}_\perp$ using features extracted from the training set.

### C.2  TOY EXAMPLE

The training data consists of $2^9$ input samples in $[0, 1]^2$ and the corresponding outputs $y = f(\mathbf{x}) + \epsilon$, where $f(\mathbf{x}) = sin(2\pi x_1)sin(2\pi x_2)$ and $\epsilon \sim \mathcal{N}(0, 0.1^2)$. OOD inputs are represented by 200 points from the Gaussian distribution $\mathcal{N}([1.5, 1.5], 0.1 * \mathbf{I})$. An additional set of 200 in-distribution points is used for validation.

The prediction network consists of an overparametrized MLP with 4 hidden layers — $MLP(2, 20, 32, 20, 2, 1)$ — and ReLu activation functions. The network is trained over 400 epochs using Adam with learning rate of $10^{-3}$, and weight decay of $10^{-4}$. We also apply batch normalization.

### C.3  HOUSING PRICE

Each network is trained to minimize the mean-squared error loss for 100 epochs using SGD with a learning rate of 0.001 and weight decay of $10^{-4}$. All networks achieve a scaled mean-squared error between 0.28 and 0.29. Based on the validation log likelihood, we select a 4-component GMM to model the DenseNet and ResNet features (see Section A.2).

### C.4  AGE ESTIMATION

All networks achieve a scaled mean-squared error between 0.18 and 0.19. Based on the validation log likelihood, we select a 2-component GMM to model the DenseNet features and a single Gaussian to model the ImageNet features (see Section A.2).

