# OpenReview forum: "Neural Network Out-of-Distribution Detection for Regression Tasks"
_ICLR.cc/2020/Conference — Reject_

### Official Review · AnonReviewer2 · 2019-10-23
**Official Blind Review #2**

**Rating:** 3

**Review:**

The authors present the empirical observation that regression models trained using weight decay tend to produce high-level features that lie on low-dimensional subspaces. They then use this observation to propose an algorithm for detecting out of distribution data by fitting a simple GMM to the features produced during training.

The main contribution of the paper is a way of calculating the "OOD scores" that are reported in the different figures. However, after reading the paper 3 times I'm still unable to find a definition of these scores. I'm guessing it's negative log-likelihood under the fitted GMM, but I'm unsure.

The observation (features on low-dimensional manifolds) is interesting, and the proposed algorithm is potentially useful. However the experiments are quite limited, with only 2 training datasets. The analysis of why/when the algorithm is expected to work and how this depends on the model is shallow. The two main results are for models pre-trained on Imagenet: presumably this has a huge impact on the features that end up being used for OOD detection, but this is not discussed in the paper at all.

Questions to the authors:
- Why focus only on regression? It seems to me that your analysis for the features lying on a low-dimensional subspace should also apply to classification models.
- How do your results depend on the specific models that are used? I can come up with models with very few final features where your method would not work. Do you have any analysis or guidance?
- What do all abbreviations in the tables mean? Please make the captions more informative so the reader does not need to search in the main text.

**Experience Assessment:**

I have published one or two papers in this area.

**Review Assessment: Checking Correctness Of Derivations And Theory:**

I assessed the sensibility of the derivations and theory.

**Review Assessment: Checking Correctness Of Experiments:**

I assessed the sensibility of the experiments.

**Review Assessment: Thoroughness In Paper Reading:**

I read the paper at least twice and used my best judgement in assessing the paper.

---

> ### Author Response · Authors · 2019-11-14
> **Author response to review**
>
> W.R.T. the generality of our method: We believe that our method is sufficiently general, as we find it successful on small CNNs (Rotated MNIST, Section 3) and larger ResNets/DenseNets (KY Houses and Adult Age Estimation, Section 6). We agree that it is possible to construct “models with very few final features where your method would not work.” Indeed, the fact that simple generative models are sufficient for OOD detection relies on the curse of dimensionality, as it is unlikely that in-distribution and OOD features occupy the same parts of a high-dimensional space. Nevertheless, most MLPs and CNNs that are commonly used in practice have 100-1000 dimensional activations, which we believe is more than sufficient for our method to work.
>
> W.R.T. application to classification problems: please see our response to Reviewer #1.
>
> W.R.T. table abbreviations: Thank you for catching that the caption is missing abbreviations; we will fix this. TNR @ TPR 95% stands for “true negative rate when using an OOD threshold that results in 95% true positive rate of detection.” AUROC stands for “area under receiver operating characteristic.”
>
> W.R.T. computing an OOD score: this score can be computed using the NLL of the GMM. We will clarify this point.

---

### Official Review · AnonReviewer3 · 2019-10-24
**Official Blind Review #3**

**Rating:** 3

**Review:**

This work is tackling the problem of doing out-of-distribution detection in regression. The core idea is to fit a simple generative model to hidden activations and use thresholded likelihood to decide whether an input is in or out of distribution. Using a simple generative model for this is motivated by the intrinsic low-dimensionality of hidden activations.

My issue with the narrative of this paper is two-fold.

First, the intrinsic low-dimensionality of the hidden activation distribution seems entirely problem-dependent. One can train networks with orthonormal layers and therefor flat spectra that work fine and there are also invertible discriminative models that do not discard any information from layer to layer, albeit being very similar to standard ResNets. It might be that this heuristic works well for the problems discussed in the paper, but I don't see any reason for this to hold in general.

Second, Nalisnick et al. don't show that deep generative models are often overconfident (as stated in the paper here). What they show is that likelihood can be higher for datasets with lower entropy than the one the deep generative model has been trained on. They also show that "graying" inputs increase the likelihood and they can even show this analytically for flow models with data-independent determinant of their Jacobian. The GMM approach proposed here is thus a special case of what Nalisnick et al. described and should have the same problem. Reducing the variance of the features should increase the likelihood.

All in all, the presented method relies too much on heuristics that seem specific to the analyzed architectures and datasets and there is no theoretical reason for this to hold in general as far as I can see.

The originality of the paper is also somewhat limited, as generative models for OOD detection are well-studied. Thresholded likelihoods are problematic in many settings, so I am not convinced this approach will work well across the board.

----

Eric Nalisnick, Akihiro Matsukawa, Yee Whye Teh, Dilan Gorur, and Balaji Lakshminarayanan. Do deep generative models know what they don’t know? In ICLR, 2019.

**Experience Assessment:**

I have published one or two papers in this area.

**Review Assessment: Checking Correctness Of Derivations And Theory:**

I carefully checked the derivations and theory.

**Review Assessment: Checking Correctness Of Experiments:**

I assessed the sensibility of the experiments.

**Review Assessment: Thoroughness In Paper Reading:**

I read the paper thoroughly.

---

> ### Author Response · Authors · 2019-11-14
> **Author response to review**
>
> W.R.T novelty of our method: we respectfully disagree that the originally our method is limited. When generative models have been applied to OOD detection, they are typically used to fit the input data distribution. In our work, we demonstrate that it is possible to achieve good OOD performance not by modelling input but instead by modelling hidden activations. We believe that this is an important contribution because input-based modelling methods cannot be applied to high-dimensional images (to the best of our knowledge), whereas our method empirically demonstrates great success on these problems.
>
> W.R.T. the “discarded information” from activations: We believe that there may have been some confusion with regards to where our method is applied. Our method ONLY looks at the activations of the last hidden layer (not the activations of other hidden layers). Information is always discarded from this layer, as these d-dimensional features are then mapped to a scalar prediction. Indeed, we specifically do not examine the activations of earlier layers because -- as you point out -- information may not be discarded and activation distributions are less likely to be intrinsically low dimensional.
>
> W.R.T. the work of Nalisnick et al. 2019: we apologize if we misstated the findings of this paper. We will revise the sentence in Section 2 that describes this work.

---

> > ### Comment · AnonReviewer3 · 2019-11-14
> > **d-dimensional features**
> >
> > Thank you for answering my concerns.
> >
> > Of course information is most likely discarded when mapping from R^d->R, but this does not tell us much about the intrinsic dimensionality of the last layers output before this is happening. And this is the space in which OOD detection is carried out if I understand correctly.
> >
> > It is possible to build invertible deep networks that preserve all information up until the final d-dimensional output and only then (when mapping from R^d->R), information is discarded. Wouldn't your assumptions break down in such a case?

---

### Official Review · AnonReviewer1 · 2019-10-30
**Official Blind Review #1**

**Rating:** 1

**Review:**

===Summary===

The authors propose to perform out of distribution detection for regression models by fitting a generative model in the feature space of the regression model. An input example is deemed to be out-of-distribution if it has low likelihood under this generative model.

===Overall Assessment===

I recommend that the paper is rejected. There are a number of aspects that need to be improved. You should fix these and resubmit to a future conference.
The paper focuses on the difference between regression and classification tasks and claims that the paper's method addresses an unmet need for OOD for regression. However, both the proposed method and the analysis justifying it are generic enough to be applied to both regression and classification.
The paper handles technical claims far too casually in sec 4 and does not provide sufficient justification that the claims are true.
There are natural baselines, such as using a generative model on the raw input space, that are ignored.

===Comments===

Remark 1 feels to me like it was added for the sake of having more math in the paper, not because it is crucial to the paper's argument.

You remark at various places that existing methods don't naturally generalize from classification to regression. However, you never fully explain why. Also, your proposed method can be applied out-of-the box to classification problems. Your analysis in sec 4 trivially applies to binary classification tasks, and could be naturally extended to multi-class classification where w is not a vector but a num_classes x num_features matrix.

The parallel should be between classification and heteroskedastic regression, since there you have a distribution per example.

The logic in "In-distribution features are intrinsically low dimensional" is insufficient

The connection between section 4 and your proposed method is not particularly precise. You also have lots of technical claims in 4 that are unsubstantiated. For example, you write "this new network will likely have less discarded information than the shallower network". What does 'likely' mean? In what sense are you making an actual technical statement? Each of the subsections in sec 4 has similar issues.
"The CNNs are pre-trained on ImageNet (Denget al., 2009) and the last layer is replaced with a linear layer that produces a single output."
Why did you do this? Did you fine tune or just retrain the top layer?

"For these two baselines, the variance of the forward passes is used as a metric for detecting OOD inputs"
   Can you explain why these are reasonable baselines for OOD? Why no baseline that fits a generative model in input space?

You should cite Ren et al. "Likelihood Ratios for Out-of-Distribution Detection"


**Experience Assessment:**

I have published one or two papers in this area.

**Review Assessment: Checking Correctness Of Derivations And Theory:**

I assessed the sensibility of the derivations and theory.

**Review Assessment: Checking Correctness Of Experiments:**

I assessed the sensibility of the experiments.

**Review Assessment: Thoroughness In Paper Reading:**

I read the paper at least twice and used my best judgement in assessing the paper.

---

> ### Author Response · Authors · 2019-11-14
> **Author response to review**
>
> Thank you for your review and for making us aware of the paper by Ren et al. We hope that we are able to clarify some misunderstandings and justify our decisions.
>
> W.R.T. claims in section 4: we believe that our language in this section may have caused some confusion. The purpose of this section was to empirically demonstrate that the final layer activations occupy a space with extremely low intrinsic dimensionality. The goal of Figure 3 and the accompanying text around it is to show that this low-dimensionality is correlated with (not necessarily caused by) increased depth and regulation. We will revise the language to clarify this.
>
>
> The purpose of Section 4 is to empirically demonstrate two phenomena that influence our method: 1) the distribution of in-distribution activations is intrinsically low-dimensional---making it possible to fit with simple generative models---and 2) OOD activations do not lie within the same distribution. We will re-arrange this section to clarify these points.
>
> W.R.T. input-space generative models as baselines: while this is a natural baseline that could be used for the toy example, we don’t believe that it is possible to apply generative models directly to the KY Houses or Adult Age Estimation datasets, as the inputs are 224 x 224 images. To the best of our knowledge, input space OOD detection has not been applied to images that are larger than 32 x 32.
>
> W.R.T. classification OOD methods not generalizing to regression: we will improve the discussion of this in Section 2. As far as we are aware, the only classification method that generalizes is the baseline of Hendrycks and Gimpel (ICLR 2017) which flags inputs as OOD if the softmax confidence falls below a threshold. This has a natural analog to heteroscedastic regression (the var method in our experiments). However, most other classification methods make use of the entire softmax distribution. For example, the ODIN method (Liang et al., ICLR 2018) applies temperature scaling to the output logits.
>
> W.R.T. experiments on classification OOD: our method can indeed be applied to classification OOD. For example, with a DenseNet-40 trained on CIFAR-10, our method can be used to detect OOD inputs from Tiny-Imagenet with an AUROC of 97% - which is comparable to many classification-specific methods. Regression was the primary focus of our work because, as we explain, there are not as many existing OOD-detection methods for this domain. Nevertheless, we can include additional classification results in the supplementary material.
>
> W.R.T. fine-tuning ImageNet networks: we fine-tune the network and the top layer, as this is common practice for datasets with images that are larger than 224 x 224.

---

### Official Review · AnonReviewer4 · 2019-10-30
**Official Blind Review #4**

**Rating:** 1

**Review:**

This paper proposes to detect inputs that are from a slightly shifted distribution (eg images of houses in CA instead of KY) or from a very shifted distribution (eg imagenet images) by fitting a density model to the last layer h(x) of an MLP trained with squared error, and using the likelhiood score p(h(x)) as a metric. This is a reasonable idea. However, it is not novel eg Aigrain'19 did essentially the same thing for classification models. (The difference between classification and regression is a trivial change to the loss function, and does not change the fundamental idea.)

In addition to lack of novelty, the experimental methodology is very weak. First, the toy 2d example is too trivial to be informative, since there is essentiallly no overlap between the two distributions of features, p(x) and q(x) - even a density model on input space could detect this. More importantly, the results on the two image datasets are suspect. First, it seems that using the predictive variance sigma(x) as the reliability metric (the "var" method) - which is totally standard approach known as 'heteroskedastic regression'. - works very well in several cases. I suspect when it fails it is due to  implementation problems (eg trying to predict sigma instead of log(sigma)). Also ensembles are known to be very robust to distribtution shift (see eg Ovadia'19), so  I am surprised at their poor performance. Another problem is that the datasets used are not standard, so it is impossible to compare to other papers. Finally, no error bars are reported, so it is hard to know if any of the results are statistically significant.



J. Aigrain and M. Detyniecki, “Detecting Adversarial Examples and Other Misclassifications in Neural Networks by Introspection,” in ICML Workshop on Uncertainty and Robustness in Deep Learning, 2019 [Online]. Available: http://arxiv.org/abs/1905.09186


Y. Ovadia et al., “Can You Trust Your Model’s Uncertainty? Evaluating Predictive Uncertainty Under Dataset Shift,” arXiv [stat.ML], 06-Jun-2019 [Online]. Available: http://arxiv.org/abs/1906.02530





**Experience Assessment:**

I have read many papers in this area.

**Review Assessment: Checking Correctness Of Derivations And Theory:**

N/A

**Review Assessment: Checking Correctness Of Experiments:**

I did not assess the experiments.

**Review Assessment: Thoroughness In Paper Reading:**

I read the paper at least twice and used my best judgement in assessing the paper.

---

> ### Author Response · Authors · 2019-11-14
> **Author response to review**
>
> W.R.T. novelty: Thank you for making us aware of Aigrain et al.’s work. However, we strongly disagree that our proposed method is “essentially the same thing.” Aigrain et al.’s method requires 1) generating adversarial inputs and artificial OOD inputs and 2) trains a neural network (a supervised method) on the logits to identify OOD inputs. Our method on the other hand 1) does not require any OOD inputs at training time (real or artificial) and 2) models the distribution of in-distribution features with a simple Gaussian Mixture Model. We believe that this makes our method substantially different and a novel contribution.
>
> W.R.T. the performance of heteroskedastic regression and ensembles: you note that these baseline methods “work very well in several cases.” Our experiments indeed confirm this: the “var” (heteroskedastic) model and the ensemble both achieve an AUROC of >90% on most of our experiments, making these methods strong baselines and not “poor” performers. At the same time, we do not find it surprising that our proposed method outperforms them. The predicted variance from a heteroskedastic model (which we did in fact parameterize logarithmically) is a reduction of the feature space down to a single dimension. Likewise, an ensemble reduces predictions down to a handful of dimensions (one for each model in the ensemble). These reductions would be prone to the curse of dimensionality argument that we outline in Section 3. Our proposed method on the other hand uses all >1000 dimensions of the feature space to determine if inputs are OOD.
>
> W.R.T the datasets: We chose to focus on image datasets which are the most common datasets for OOD and domain adaptation datasets. This is likely because it is easy to generate images which are “out of distribution” (i.e. use images from an unrelated dataset), whereas it would be more difficult to generate OOD data for e.g. tabular datasets.
>
> We introduced new datasets because we were not aware of any image datasets used for regression. We view this as a contribution in its own right, as future regression-OOD research can utilize these datasets/baselines for evaluation.
>
> W.R.T. error bars: we will update our results table with multiple runs of our experiments.

---

### Decision · Program_Chairs · 2019-12-19

**Decision:**

Reject

**Comment:**

The paper investigates out-of-distribution detection for regression tasks.

The reviewers raised several concerns about novelty of the method relative to existing methods, motivation & theoretical justification and clarity of the presentation  (in particular, the discussion around regression vs classification).

I encourage the authors to revise the draft based on the reviewers’ feedback and resubmit to a different venue.